# Interstitial HDR Brachytherapy in the Treatment of Non-Melanocytic Skin Cancers around the Eye

**DOI:** 10.3390/cancers13061425

**Published:** 2021-03-20

**Authors:** Paweł Cisek, Dariusz Kieszko, Mateusz Bilski, Radomir Dębicki, Ewelina Grywalska, Rafał Hrynkiewicz, Dominika Bębnowska, Izabela Kordzińska-Cisek, Agnieszka Rolińska, Paulina Niedźwiedzka-Rystwej, Ludmiła Grzybowska-Szatkowska

**Affiliations:** 1Department of Brachytherapy, St. John’s Cancer Center, 20-090 Lublin, Poland; pcisek@interia.eu (P.C.); dkieszko@cozl.pl (D.K.); rdebicki@cozl.pl (R.D.); 2Department of Radiotherapy, Medical University of Lublin, 20-093 Lublin, Poland; ludmila.grzybowska-szatkowska@umlub.pl; 3Department of Radiotherapy, St. John’s Cancer Center, 20-090 Lublin, Poland; 4Department of Clinical Immunology and Immunotherapy, Medical University of Lublin, 20-093 Lublin, Poland; ewelina.grywalska@umlub.pl; 5Department of Clinical Immunology, St. John’s Cancer Center, 20-090 Lublin, Poland; 6Institute of Biology, University of Szczecin, Felczaka 3c, 71-412 Szczecin, Poland; rafal.hrynkiewicz@usz.edu.pl (R.H.); dominika.bebnowska@usz.edu.pl (D.B.); 7I Department of Clinical Oncology, St. John’s Cancer Center, 20-090 Lublin, Poland; ikordzinska@cozl.pl; 8Department of Applied Psychology, Medical University of Lublin, 20-093 Lublin, Poland; aga.rolinska@wp.pl

**Keywords:** HDR brachytherapy, non-melanocytic skin cancer, cancer around the eye, radiation therapy

## Abstract

**Simple Summary:**

Eyelid tumors account for approximately 3% of all head and neck cancers and 5 to 10% of all skin cancers. Among the basic methods of treating eyelid tumors, apart from surgery, is radiotherapy, but this method carries a high risk of complications within the eye lens and may lead to the development of cataracts. Interstitial HDR brachytherapy is a less invasive method of skin cancer treatment. Unfortunately, the analysis of the literature to date has shown that it is rarely used in the treatment of skin cancer in this location. In our study, we analyzed the results of 28 patients treated with HDR interstitial brachytherapy. We showed that this is a highly effective, short-lived and relatively low burden method of treating patients with neoplasms of the skin of the eyelids, medial and lateral angles, and skin cancer of the cheek, nose and temples with an infiltration of the ocular structures.

**Abstract:**

Background: Eyelid tumors are rare skin cancers, the most common of which is basal cell carcinoma characterized primarily by local growth. In addition to surgery, radiotherapy is among the basic methods of treatment. External beam radiotherapy is associated with the risk of complications within ocular structures, especially the lens. In the case of interstitial brachytherapy, it is possible to administer a high dose to the clinical target volume (CTV), while reducing it in the most sensitive structures. Methods: This paper presents the results of an analysis of 28 patients treated with interstitial high dose rate (HDR) brachytherapy for skin cancers of the upper and lower eyelid; medial and lateral canthus; and the cheek, nose and temples with the infiltration of ocular structures. The patients were treated according to two irradiation schedules: 49 Gy in 14 fractions of 3.5 Gy twice a day for 7 days of treatment, and 45 Gy in 5 Gy fractions twice a day for 5 days. The mean follow-up was 22 months (3–49 months). Results: two patients (6%) had a relapse: a local recurrence within the irradiated area in one of them, and metastases to lymph nodes in the other. The most common early complication was conjunctivitis (74%), and the most common late complication was dry eye syndrome (59%). Conclusions: Interstitial HDR brachytherapy for skin cancers of the upper and lower eyelid; medial and lateral cants; and the cheek, nose and temples with infiltration of ocular structures is a highly effective, short and relatively low burden type of treatment.

## 1. Introduction

Eyelid tumors are rare skin cancers, accounting for about 3% of all head and neck cancers and 5–10% of all skin cancers. Annual incidence is only 1.37 per 100,000 people. Basal cell carcinoma is the most common histopathological type, accounting for about 90% of cases. Squamous cell carcinoma, sebaceous gland carcinoma, Merkel cell carcinoma and melanoma are less common [1,2]. While basal cell carcinoma is primarily characterized by local growth and low metastatic potential (0.028 to 0.55%), squamous cell carcinoma is an aggressive cancer with metastasis percentages between 5 and 12% [1,3,4]. Squamous cell carcinoma also has a worse prognosis, with 5- and 10-year survival rates at 90 and 83–87%, respectively. In basal cell carcinoma, 5- and 10-year survival rates are close to 100% [4]. Assessment by the general practitioners is often a key step towards a patient diagnosis. To help physicians in selecting skin cancer patients, a checklist was developed that provides essential tips to accelerate the patient’s progress throughout the diagnosis stage and thus increase the chances of recovery. Unfortunately, the further stages of cancer diagnosis can also be difficult [5]. Despite the fact that these two types of carcinoma are well characterized, they are often misdiagnosed in dermatoscopic diagnosis. In order to avoid basal cell carcinoma being incorrectly defined as squamous cell carcinoma and vice versa, the criteria for its differentiation must be suitably critical [6]. The most common location within the eye is the lower eyelid (50–60%) and medial canthus (25–30%) [1]. The basic method of treatment is surgery; the others are cryotherapy, laser therapy, local chemotherapy, photodynamic therapy, immunotherapy and radiation therapy [7,8]. However, some of these methods may not cause acceptable side effects in patients after treatment. For patients with carcinoma of the eyelids or lips who do not agree to the proposed methods of treatment for aesthetic and functional reasons, radical radiotherapy remains a good solution as the basic therapy [9]. Commonly used radiotherapy techniques include X-ray teleradiotherapy, megavoltage photon teleradiotherapy, electron teleradiotherapy and brachytherapy, but Hedgehog pathway inhibitors may also be an effective and attractive therapy for local basal cell carcinoma located in the eyes and eyelids [10].

The main limitation of the use of radiation therapy is its toxicity to ocular structures, primarily to lenses, which, due to their low radiation tolerance, are exposed to late toxicity consisting of lens fibrosis and, consequently, cataracts [11]. The advantage of interstitial brachytherapy is the possibility of placing applicators inside the target, which makes it possible to administer a high dose within the clinical target volume (CTV). The rapid decrease in the dose with the growing distance from the applicator allows for the protection of adjacent structures, in particular, for a significant dose reduction within the most sensitive structures—the lenses.

## 2. Materials and Methods

### 2.1. Group Characteristics

The analysis included 28 patients diagnosed with skin cancers of the upper and lower eyelid; medial and lateral canthus; and the cheek, nose and temples with infiltration of the above-mentioned ocular structures. The patients were treated with HDR brachytherapy at the Brachytherapy Department of the Centre of Oncology of the Lublin Region between 2012 and 2017. All patients received radical treatment. Brachytherapy was used as independent treatment in 24 patients, and as adjuvant treatment after surgery in the other 4 patients. The indication for treatment was the microscopic non-radicality of the procedure. In the case of brachytherapy as an independent treatment, 8 patients had undergone surgery and suffered relapse after the treatment, and for the remaining patients, brachytherapy was the first method of cancer treatment. None of the patients had previously received any other radiation therapy. None of the patients were clinically diagnosed with lymph node metastases (9 patients had previously had a neck ultrasound, and 1 patient had had a CT scan of the neck). No distant metastases were found in any of the patients (all patients had chest X-ray, and 4 of them had abdominal ultrasound as well). The clinical and histopathological characteristics are presented in Table 1. 

### 2.2. Application Procedure

The application procedure in most patients (26) was performed under local infiltration anesthesia with 1% lignocaine. In the remaining patients, the procedure was performed under short intravenous anesthesia with propofol and fentanyl. Flexible applicators of 35 cm long made by Varian Medical System were used. They were inserted under the skin at a depth of 2–5 mm, parallel to each other, according to the Paris system, so that they covered the tumor or the tumor bed with a 1–2 cm healthy skin margin. The applicators were placed densely, every 2–6 mm, which allowed for a rapid decrease in the dose outside the irradiation area (Figure 1). After the application procedure, a tomography for treatment planning was performed using a 32-row Siemens CT scanner. The slice thickness was 1–3 mm. GTV was defined as the area with clinically evident neoplastic infiltration, and CTV comprised the infiltrate with a healthy tissue margin of 0.5–1 cm. In the case that the infiltration border was the eyeball, the CTV did not include the eyeball structures. Based on the location of the applicators, the CTV area was contoured on tomography scans for treatment planning. The eyeball, lens, optic nerve and retina were also drawn, as well as the structure of the opposite eye, if relevant to the location of the tumor.

### 2.3. Application Procedure

The patients were treated according to 2 irradiation schedules: 49 Gy in 14 fractions of 3.5 Gy twice a day for 7 days of treatment (9 patients) and 45 Gy in 5 Gy fractions twice a day for 5 days (19 patients). The choice of schedule was determined by a number of factors. The shorter one was used in the elderly, patients in worse general condition and patients with less advanced disease. Treatment planning was carried out using the BrachyVision treatment planning system. An Ir 192 source with a diameter of 0.6 mm and an average activity of 10 Ci was used. Treatment was performed using a 24-channel GammaMed or GammaMedplus (Varian) afterloader.

### 2.4. Post-Treatment Surveillance

In the post-treatment period, patients were subjected to periodic clinical evaluation, initially every month for the first 3 months; some of them were also subjected to ophthalmic evaluation. Ultrasound of the neck and abdomen and chest X-ray were also carried out periodically in selected patients. Patients were evaluated for local recurrence of lymph node metastases and distant metastases. The frequency and severity of early and late side-effects of radiation were assessed as well. The Common Terminology Criteria for Adverse Events (CTCAE) v 4.0 scale and the Radiation Therapy Oncology Group (RTOG) scale were used to assess toxicity [12,13].

## 3. Results

### 3.1. Dosage and Treatment Planning

In patients with a fraction dose of 3.5 Gy, the average D90 (dose in 90% isodose) was 3.9 ± 0.4 Gy. The median D90 was 3.7 Gy (3.5–4.3 Gy). The mean D100% (dose in 100% isodose) was 2.6 ± 0.6 Gy. The median D100% was 2.7 Gy (2–3.2 Gy). The mean volume of irradiated lesion was 8.0 ± 6.7 cm^3^. In patients with a fraction dose of 5 Gy, the mean D90 was 5.4 ± 0.5 Gy. The median D90 was 5.5 Gy (5–6 Gy). The mean D100% (dose in 100% isodose) was 3.7 ± 0.8 Gy. The median D100% was 3.7 Gy (3–4.6 Gy). The mean volume of irradiated lesion was 8.1 ± 6.7 cm^3^. The total doses in both fractionation schemes were converted into the biologically effective dose (BED) and the 2 Gy-per-fraction equivalent dose (EQD2) for alpha/beta 10. The data are presented in Table 2.

The median volume of irradiated lesion was 8.8 cm^3^ (1.4–14.8 cm^3^)—in patients treated with a 3.5 Gy dose, it was 10.5 (4.3–13.1) cm^3^, and in patients treated with dose of 5 Gy, it was 6.6 (1.4–8.9) cm^3^. 

Doses in critical organs were reported for the maximum dose in the nearest lens, eyeball, retina and optic nerve. All doses were converted to BED and EQD2 for alpha/beta 3. The data are presented in Table 3.

During the mean follow-up of 24 months (4–49 months), two patients (7%) had a relapse. In one of them, it was a local recurrence in the irradiated area. This was a patient with basal cell carcinoma of the canthus, and the recurrence, which was confirmed histopathologically, took place during the 8th month of follow-up. The second patient had metastases to submandibular lymph nodes after 3 months. The patient had extensive cutaneous squamous cell carcinoma of the cheek that covered the lower eyelid.

### 3.2. Toxicity

The early and late toxicity of the radiation therapy and application procedure was analyzed. Complications from the application procedure and acute toxicity were assessed in all 28 patients, and late toxicity was assessed in 20 patients. The data are presented in Table 4.

## 4. Discussion

An analysis of the available literature indicates the rarity of the use of interstitial HDR brachytherapy in the treatment of skin cancers in this location. Available analyses regarding HDR brachytherapy involve small groups of patients (8–20 patients) [14,15,16,17,18], and LDR brachytherapy is more common (20–160 patients) [19,20,21,22,23]. Studies indicate the high efficiency of brachytherapy, in the range of 94–100%, both among patients treated with HDR brachytherapy and LDR brachytherapy [14,15,16,17,18,19,20,21,22,23]. A study by Vavssori et al. included 10 patients with eyelid tumors using contact high dose rate brachytherapy (HDR-BT) with a customized applicator. Except for one patient, all patients were perfectly tolerant to the treatment. After a median follow-up time of 51 months, none of the patients experienced optic neuropathy, retinopathy, lacrimation or visual impairment. Moreover, no relapse or metastasis was observed in patients, and the cosmetic effects were satisfactory. These data indicate that this method is effective and safe in the treatment of eyelid s and may be an effective alternative to other therapies [18]. There are also reports indicating that the therapeutic and cosmetic effect of brachytherapy in this localization may be unfavorable [24]. In the analyzed group of patients, the local control throughout the entire follow-up period was also high and amounted to 97%. These results are similar to those of teleradiotherapy (93–96.5%) [23,25,26] or surgical treatment (80–90%) [27,28,29]. Meta-analysis covering nearly 10,000 patients comparing EBRT vs. BTR showed a similar efficacy of both treatment methods in non-melanocytic skin tumors with a better cosmetic effect of brachytherapy [30].

Doses in the analyzed group of patients were reported in those ocular structures that are the most sensitive to radiation. Due to the properties of the source and the dense arrangement of applicators, a sharp dose reduction outside the irradiation area was achieved. The median total dose in the organs analyzed for the 3.5 Gy/49 Gy/BID scheme was 18.2, 12.6, 48.6 and 10.8 Gy in the lens, retina, cornea and optic nerve, respectively, and 17.1, 11.7, 44, 1 and 9 Gy. Studies show that the lens is the most sensitive to radiation [11]. Although a late radiation side-effect in the form of a cataract can occur even after a single dose of radiation of 2 Gy, it does not affect the quality of vision [11]. The severity and timing of cataracts are dose dependent. Studies indicate that irradiation up to a dose in the 2.5–6.5 Gy range is associated with a 33% cataract risk within 8 years, and irradiation up to a dose in the 6.5–11.5 Gy range with a cataract risk of 66% within 4 years [31]. Emami et al. [32] showed that there is a 5% risk of cataracts in 5 years (TD5/5) for 10 Gy, and a risk of 50% in 5 years (TD50/5) for 18 Gy. The risk of cataracts is associated with a number of factors, such as age, type of radiation and fraction dose [33,34,35]. Most of the patients in the analyzed group were elderly. Although the risk of radiation cataracts is lower in older patients than in younger patients irradiated with the same dose, radiation-induced cataracts in the elderly are concurrent with cataracts caused by other factors [33]. Research also indicates that the risk of cataracts increases with a higher fraction dose and is greater for brachytherapy and electron radiation than for photon radiation [35]. In the analyzed group of patients, the use of brachytherapy was associated with a short hospitalization time and a relatively low burden on the patient, although a slightly longer 1.5-week treatment regimen due to a lower fraction dose may be more beneficial in terms of lens protection. 

Research indicates that the risk of retinal damage occurs when the 24 Gy dose is exceeded. TD5/5 is 45–50 Gy, and TD50/5 is 55 Gy [32,36]. The tolerance dose was not exceeded in the analyzed group of patients. Similarly, the optic nerve tolerance dose was not exceeded. In the study by Emami et al. [32] and analysis of Mayo et al. [37], the researchers indicate that a dose lower than 50 Gy is associated with a risk of nerve damage of less than 5%. Despite a relatively high dose in the cornea, resulting from the close proximity of applicators, studies indicate that the risk of corneal damage occurs after exceeding the 50 Gy dose, and the risk of corneal ulceration after exceeding 60 Gy [37,38].

## 5. Conclusions

Interstitial HDR brachytherapy for skin cancers of the upper and lower eyelid; medial and lateral cants; and the cheek, nose and temples with infiltration of ocular structures is a highly effective, short and relatively low burden type of treatment. The profile of expected toxicity is favorable. No severe and late CTCAE ≥ 3 or late RTOG ≥ 3 toxicity was observed.

## Figures and Tables

**Figure 1 cancers-13-01425-f001:**
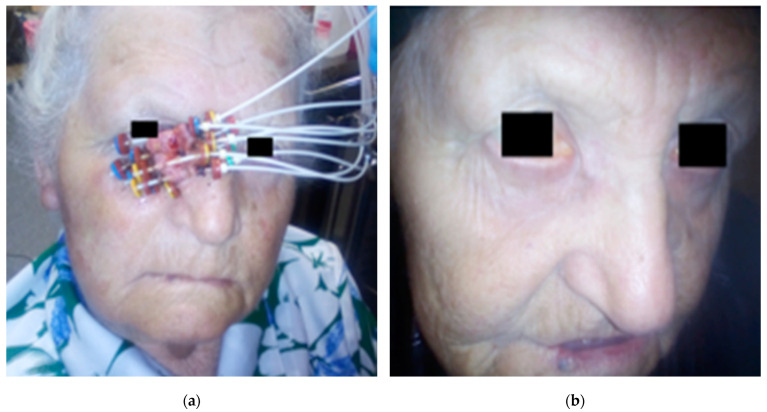
(**a**) Patient, age 80, basal cell carcinoma of the lower eyelid and medial canthus of the right eye. Applicators placed in the tumor; (**b**) the same patient, six months after the end of treatment.

**Table 1 cancers-13-01425-t001:** The clinical and histopathological characteristics of patients.

Clinical and Histopathological Factors	Number/Median (Range)
age	82 (64–96)
Sex	
Men	17
Women	11
Histopathological type	
Basal cell carcinoma	24
G1 squamous cell carcinoma	2
G2 squamous cell carcinoma	2
Stage	
T1N0M0	23
T2N0M0	5
Location	
Lower eyelid	14
Upper eyelid	1
Medial canthus	11
Lateral canthus	1
Lower eyelid and medial canthus	1
Number of applicators	5 (3–11)

**Table 2 cancers-13-01425-t002:** Characteristics of treatment schemes and dosage.

Fractionation Scheme		Median	Mean	Min	Max
3.5 Gy/49 Gy/BID (two times a day)	D90	51.8	54.6	49.0	60.2
	BED D90	71.0	75.9	66.2	86.1
	EQD2 D90	59.1	63.3	55.1	71.7
	D100	37.8	36.4	28.0	44.8
	BED D100	48.0	45.9	33.6	59.1
	EQD2 D100	40.0	38.2	28.0	49.3
5 Gy/45 Gy/BID	D90	48.6	49.5	45.0	54.0
	BED D90	74.8	76.7	67.5	86.4
	EQD2 D90	62.4	63.9	56.3	72.0
	D100	33.3	34.2	27.0	41.4
	BED D100	45.6	47.2	35.1	60.4
	EQD2 D100	38.02	39.3	29.3	50.4

**Table 3 cancers-13-01425-t003:** Characteristics of doses achieved in critical organs.

Fractionation Scheme		Median	Mean	Min	Max
3.5 Gy/49 Gy/BID	Dmax lens	18.2	19.1	9.8	32.4
	BED Dmax	25.5	28.8	12.1	55.7
	EQD2 Dmax	15.3	17.3	7.3	33.4
	Dmax eyeball	48.6	48.9	43.4	57.8
	Dmax retina	12.6	11.2	7.0	24.9
	Dmax nerve	10.8	9.8	5.6	22.1
5 Gy/45 Gy/BID	Dmax lens	17.1	17.2	9.9	34.2
	BED Dmax	27.9	28.2	13.5	77.5
	EQD2 Dmax	16.8	16.9	8.2	46.5
	Dmax eyeball	44.1	45.2	36.9	53.1
	Dmax retina	11.7	12.9	9.9	35.1
	Dmax nerve	9.0	9.8	3.6	24.3

**Table 4 cancers-13-01425-t004:** Characteristics of early and late toxicity of radiation therapy and application procedure. CTCAE—Common Terminology Criteria for Adverse Events; RTOG—Toxicity criteria of the Radiation Therapy Oncology Group.

**Complications from the Application Procedure**	**CTCAE/number (Percentage) of Patients**	**RTOG/number (Percentage) of Patients**
oedema	CTCAE 1–28 (100%)	-
hematoma	CTCAE 1–3 (11%)	-
**Early complications**	**CTCAE/number (percentage) of patients**	**RTOG/number (percentage) of patients**
skin	-	RTOG 1–18 (64%)
		RTOG 2–9 (32%)
		RTOG 3–1 (3%)
Conjunctivitis	CTCAE 1–20 (71%)	-
	CTCAE 2–1 (3%)	
**Late complications**	**CTCA/number (percentage) of patients**	**RTOG/number (percentage) of patients**
Eyelid deformity	CTCAE 1–6 (30%)	-
	CTCAE 2–1 (5%)	
Dry eye syndrome	CTCAE 1–11 (55%)	-
Skin lesions (discoloration, thinning, telangiectasia)	-	RTOG 1–16 (80%)

## Data Availability

Due to privacy and ethical concerns, the data that support the findings of this study are available on request from the corresponding author, M.B.

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
