# Peer review of "Interstitial HDR Brachytherapy in the Treatment of Non-Melanocytic Skin Cancers around the Eye"

_cancers, 2021, doi:10.3390/cancers13061425_

Round 1

Reviewer 1 Report

In this article, the authors have analyzed the results of patients treated with HDR interstitial brachytherapy for ocular tumors. They found this method to be highly effective, short-lived and relatively less burdensome. I have some comments/concerns:

  1. Did the patients under treatment had any comorbidities?
  2. Did the therapy cause immunosuppression? 
  3. Please cite the study by Vassori et al. J Contemp Brachytherapy. (2019) 11:443-448.

Author Response

Dear Editors,

      On behalf of the Authors of our review paper entitled “Interstitial HDR brachytherapy in the treatment of non-melanocytic skin cancers around the eye” (Cancers-1137354) we would like to cordially thank for your review. We tried to follow all your suggestions and corrected the manuscript accordingly. Here are the point-by-point answers, all changes in the manuscript are marked in red.

Did the patients under treatment had any comorbidities?

RE: The patients taken into the account in the study had no comorbidities.

Did the therapy cause immunosuppression? 

RE: The therapy did not cause immunosuppression.

Please cite the study by Vassori et al. J Contemp Brachytherapy. (2019) 11:443-448.

RE: The suggested paper has been cited, thank you for your valuable comment.

Again, we would like to thank you for your valuable comments to improve our manuscript. We are hoping that in its current form the manuscript will fulfill the requirements of Cancers.

On behalf of the Authors,

Paulina Niedźwiedzka-Rystwej

Reviewer 2 Report

The authors deal in this paper with a very troublesome and interesting topic in oncology because the anatomical site of the BCC is around the eye.

The paper is well written however I have some points which I would like the authors to consider in order to improve the scientific soundness of their work:

1) Compared to most series previously published on the topic the number of patients included is somehow remarkable. For this reason in my view there is no need to include a few more patients whose primary tumor is not from eyelid but from the surronding tissues (cheek, nose and temples). I think the scientifc validity of the results would be more robust if these few patients were excluded from the statistical analysis.

2) It is not clear to me the sentence "In the case of brachytherapy as independent treatment, nine patients had undergone surgery before". if these patients had undergone surgery they should be considered as adjuvant or if they have recurred it should be clearly stated and also modificed accordigly in table 1 (rT1, rT2 etc.).

3) It is important to define more clearly the iamging modality used (and in how many patients) to define the local stage consequenlty the GTV.

4) If available a picture of the primary lesion shown in figure 1 before the implant would be a great addition fot the readers.  

5) In the discussion section authors should argue about the chance to use mould-based surface high-dose rate brachytherapy for eyelid carcinoma which has been recently investigated by other groups with promising results and potentially identify different indications for the two types of apporach.

Author Response

Dear Reviewer,

      On behalf of the Authors of our review paper entitled “Interstitial HDR brachytherapy in the treatment of non-melanocytic skin cancers around the eye” (Cancers-1137354) we would like to cordially thank for your review. We tried to follow all your suggestions and corrected the manuscript accordingly. Here are the point-by-point answers, all changes in the manuscript are marked in red.

  • Compared to most series previously published on the topic the number of patients included is somehow remarkable. For this reason in my view there is no need to include a few more patients whose primary tumor is not from eyelid but from the surronding tissues (cheek, nose and temples). I think the scientifc validity of the results would be more robust if these few patients were excluded from the statistical analysis.

RE: As suggested by the Reviewer, the patients with the primary tumor from the surrounding tissues were removed, and the statistical analysis has been adjusted.

  • It is not clear to me the sentence "In the case of brachytherapy as independent treatment, nine patients had undergone surgery before". if these patients had undergone surgery they should be considered as adjuvant or if they have recurred it should be clearly stated and also modificed accordigly in table 1 (rT1, rT2 etc.).

RE: In patients undergoing brachytherapy, 9 of them had previously underwent surgery. These were relapses after the applied treatment. There were no indications for adjuvant treatment immediately after surgery. This is explained in the text in the current form of the manuscript.

  • It is important to define more clearly the iamging modality used (and in how many patients) to define the local stage consequenlty the GTV.

RE: Due to low tumor advancement, GTV was determined on the basis of clinical evaluation. This has been supplemented in the text.

  • If available a picture of the primary lesion shown in figure 1 before the implant would be a great addition fot the readers.  

RE: Unfortunately, the picture of the primary lesion is unavailable.

  • In the discussion section authors should argue about the chance to use mould-based surface high-dose rate brachytherapy for eyelid carcinoma which has been recently investigated by other groups with promising results and potentially identify different indications for the two types of apporach.

RE: A correction was made to the discussion section. Searching the Pubmed database found no articles using brachytherapy in these cases.

Again, we would like to thank you for your valuable comments to improve our manuscript. We are hoping that in its current form the manuscript will fulfill the requirements of Cancers.

On behalf of the Authors,

Paulina Niedźwiedzka-Rystwej

Round 2

Reviewer 2 Report

The authors have properly addressed almost oll of the the points previously highlighted. I would jsut like the authors to evalute including within the discussion the results of this paper

"Vavassori A, Riva G, Durante S, Fodor C, Comi S, Cambria R, Cattani F, Spadola G, Orecchia R, Jereczek-Fossa BA. Mould-based surface high-dose-rate brachytherapy for eyelid carcinoma. J Contemp Brachytherapy. 2019 Oct;11(5):443-448. doi:10.5114/jcb.2019.88619".

I have no further comments.

Author Response

Dear Reviewer,

Thank you for your valuable comments. The paper proposed by you has been added to Discussion section and marked in green.

We hope this addition will fulfill your requirements. 

Best regards,

Paulina Niedźwiedzka-Rystwej